# Barriers and facilitators to community acceptability of integrating point-of-care testing to screen for sickle cell disease in children in primary healthcare settings in rural Upper East Region of Northern Ghana

**Samuel T. Chatio**[1]*, **Enyonam Duah**[1], **Lucy O. Ababio**[1], **Nicola Lister**[2], **Olufolake Egbujo**[2], **Kwaku Marfo**[2], **Raymond Aborigo**[1], **Patrick Ansah**[1], **Isaac Odame**[3]

1 Navrongo Health Research Center, Navrongo, Ghana, 2 Global Health & Sustainability, Novartis Pharma AG, Basel, Switzerland, 3 Hemoglobinopathy Program, The Hospital For Sick Children, University of Toronto, Toronto, ON, Canada

* schatio@yahoo.co.uk

## Abstract

### Introduction

Sickle cell disease (SCD) remains a public health problem especially in sub-Saharan Africa including Ghana. While pilot initiatives in Africa have demonstrated that neonatal screening coupled with early intervention reduces SCD-related morbidity and mortality, only 50–70% of screen-positive babies have been successfully retrieved to benefit from these interventions. Point-of-care testing (POCT) with high specificity and sensitivity for SCD screening can be integrated into existing immunization programs in Africa to improve retrieval rates. This study explored community acceptability of integrating POCT to screen for SCD in children under 5 years of age in primary healthcare facilities in Northern Ghana.

### Method

This was an exploratory study using qualitative research approach where 10 focus group discussions and 20 in-depth interviews were conducted with community members and health workers between April and June 2022. The recorded interviews were transcribed verbatim after repeatedly listening to the recordings. Data was coded into themes using QSR Nvivo 12 software before thematic analysis.

### Results

Most participants (70.9%) described SCD as serious and potentially life-threatening condition affecting children in the area. Of 148 community members and health workers, 141 (95.2%) said the screening exercise could facilitate diagnosis of SCD in children for early management. However, discrimination, fear of being tested positive, stigmatization, negative health worker attitude linked with issues of maintaining confidentiality were reported by

**Data Availability Statement:** All relevant data are within the paper and its Supporting Information files.

**Funding:** This study was part of a research collaboration grant provided by Novartis Pharma AG (grant ID: CGZX411A12402R) to Hospital for Sick Children and Navrongo Health Research Center.

**Competing interests:** The authors have declared that no competing interest exist.

participants as key factors that could affect uptake of the SCD screening exercise. Most participants suggested that intensive health education (78.3%), positive attitude of health workers (69.5%), and screening health workers not being biased (58.8%) could promote community acceptability.

## Conclusion

A large majority of participants viewed screening of SCD in children as very important. However, opinions expressed by most participants suggest that health education and professionalism of health workers in keeping patients' information confidential could improve the uptake of the exercise.

## Introduction

Sickle cell disease (SCD) remains a public health problem in the world especially people around malaria-endemic areas [1,2]. Sub-Saharan Africa shoulders the heaviest burden with about 300,000 babies born annually with SCD-SS, which is the homozygous form [3]. For instance, the frequency of sickle cell trait is 15% to 30% in west African countries such as Ghana and Nigeria [4,5]. In Ghana, 2%, translating to about 15,000 of newborns are diagnosed to have Sickle Cell Disease annually [6,7].

Sickle cell disease is inherited as an autosomal recessive disorder; affected people inherit two variant hemoglobin genes, both of which could be the gene for hemoglobin S (SS) or a combination of combination of S and another variant such as hemoglobin C (SC). A person could have sickle cell trait when the hemoglobin S gene is inherited from only one parent and a normal hemoglobin gene (A) is inherited from the other parent [8].

The World Health Organization (WHO) has estimated that SCD contributes to 9–16% of under 5 mortality, in certain regions of West Africa [9]. Further, an estimated 50–90% of babies born with SCD die before their fifth birthday [10,11]. However, there is a severe dearth of resources for diagnosis and treatment of patients with SCD in these regions. Several pilot initiatives in Africa have demonstrated that newborn screening coupled with early intervention for SCD is feasible and can yield outcomes commensurate with those achieved in high-income countries [12]. However, there is no sub-Saharan African country that has been able to successfully implement universal new-born screening for SCD as a public health program.

While pilot initiatives in Africa coupled with early intervention reduces SCD-related morbidity and mortality, only 50–70% of screen-positive children have been successfully retrieved to benefit from these public health interventions [9,13,14]. To address this problem, innovation is needed to bridge the diagnostic gap and develop sustainable healthcare programs through universal health coverage. Point-of-care testing (POCT) with high specificity and sensitivity for SCD screening can be more easily integrated into existing immunization programs in Africa and are more likely to improve retrieval rates [15]. However, previous studies demonstrated that acceptability and uptake of health interventions at the community level is key for a successful implementation of these interventions to improve health outcomes [16,17]. Therefore, this study explored community perceptions and acceptability of integrating Gazelle POCT to screen for SCD in children under 5 years of age in primary healthcare facilities to initiate care and referral pathways for early and comprehensive management in rural **Kassena-Nankana Districts in the Uppers East Region of** Northern Ghana.

## Methods/materials

### Ethical considerations

The study protocol was approved by Ghana Health Service Ethics Review Committee (**GHS-ERC:015/03/22**) and the Navrongo Health Research Centre Institutional Review Board (**NHRCIRB451**). Written informed consent was obtained from all participants before they were interviewed. Informed consent was obtained without undue influence or misrepresentation of the potential benefits and risks that might be associated with participation in the study. Prior to participation in the study, all participants were told about the purpose and study procedures. The authors had no access to information that would enable them identify study participants.

### Study design

This was an exploratory study using qualitative research approach where 16 focus group discussions (FGDs) and 29 in-depth interviews (IDIs) were conducted with community members and health workers between April and June 2022 to obtain insights into the feasibility and community acceptability of integrating Gazelle POCT technology to screen for sickle cell disease in children under 5 years of age in primary healthcare facilities in the study area.

### Gazelle point-of-care testing technology

Gazelle is a single-use cartridge-based test that enables rapid, affordable, accurate diagnosis for both SCD and sickle cell trait at the point of care. It is a cellulose acetate-based microchip electrophoresis system within a portable instrument that applies the principle of standard electrophoresis method with inbuilt analysis software, electronic data storage, and wireless data transmission capabilities. Subjects with SCD can be identified at 100% sensitivity and specificity from normal subjects and subjects with sickle cell trait. Additionally, subjects with sickle trait can be identified at 98.1% sensitivity and 97.0% specificity from normal subjects. The device is operated by rechargeable lithium batteries that can test all day on a single charge. The device also carries advanced features such as WiFi, GPS, and Bluetooth that allows connectivity for easy tracking of samples [18,19].

### Study site

The study was implemented in the Kassena-Nankana East Municipality (KNEM) and Kasena-Nankana West District (KNWD) in the upper East Region of Northern Ghana using the Navrongo Health Research Centre's (NHRC) Platform. The two districts cover an area of 1,675 square kilometres with an estimated population of about 162,000 under surveillance by the Navrongo Health and Demographic Surveillance System (NHDSS) operating under the Navrongo Health Research Centre. The population is predominantly rural with subsistence farming as the mainstay of the districts' economy. The districts have two referral hospitals located in the capital towns of Navrongo in the KNEM and Paga in the KNWD that serves as a referral point for all the health facilities in the area. The study site has different types of health providers operated and managed by the government and private individuals. There are also Community-Based Health Volunteers in the communities providing basic healthcare services to community members through the supervision of professional healthcare workers.

### Theoretical framework for the study

Theories help to explain human behaviours and are therefore very important in designing and implementing research that seeks to address a public health problem [20]. The theoretical

framework on acceptability of healthcare interventions validated by Sekhon et al., was adopted and used in this study [21]. The framework identifies various factors such as individual attitude, burden of health program, ethicality, intervention coherence, opportunity cost/benefits, perceived effectiveness of intervention and individual self-efficacy may influence community interest, acceptability and uptake of public health interventions introduced to improve health outcomes at the community level. Further, the theory identifies three components of acceptability including prospective acceptability (i.e prior to participating in the intervention, concurrent acceptability (i.e whilst participating in the intervention and retrospective acceptability (i.e after participating in the intervention).

## Study population

The study population included male and female community members (i.e mothers with children less than 5 years of age, opinion leaders) and health professionals in the study area. Mothers and opinion leaders took part in the FGDs while the IDIs were conducted with parents of children with sickle cell condition and health professionals such as Newborn and CHPS Coordinators, Community Health Nurses, Medical Assistants, Public Health Nurses as well as District and regional Directors of Health Services.

## Sampling procedures

Purposive sampling technique was used in this study to select communities, health facilities and participants for the interviews. The NHDSS is divided into five (5) zones (i.e Central, North, East, West and South) described as strata. The East and South zones are Nankani speaking while the West and North zones are Kasem speaking. The Central zone is cosmopolitan, which was not included in the sampling process. Using these demarcations, we purposively selected 4 high delivery Community-based Health Planning and Services (CHPS), clinics and health centres in each of the language groupings (i.e. Kasem and Nankani) for the study. Thus, health workers such as medical assistants and community health nurses in the study health facilities were purposively selected and invited for the interviews. The other interviews with Directors, Public health nurses and newborn and CHPS coordinators were conducted at the district and regional levels respectively. These category of health professionals were also purposively selected and included in the study due to their role in policy implementation at the district and regional levels.

Similarly, purposive sampling technique was used to select four communities (i.e. Wuru, Chiana-Asunia representing the Kasem speaking area and Mirigu, Biu representing the Nankani speaking area) within the catchment areas of the study health facilities for the FGDs with community members. Thus, with support from community health volunteers working in the selected communities we purposively selected community members such as opinion leaders and mothers of children less than five years to participate in the FGDs. Before the study, there was already SCD clinic at the War Memorial Hospital to take care of children with SCD in the area. Therefore, list of these children was obtained and using the list, data collectors visited parents/caretakers of these children and invited them into the study to share their views and experience about sickle cell disease. Community members and health professionals also shared their views on the feasibility and acceptability of using Gazelle POCT technology to screen for sickle cell disease in children under 5 years of age in the study area.

Prior to launching data collection, the study team conducted recognizance visits to all the selected health facilities, the two district health management teams and the regional health management team to inform them about the study and their permission and support to implement the activities of the study in the selected health facilities. Introductory letters were

obtained to the districts and health facilities from the regional Director of Health Services to interact with the district and facility leaders to understand the dynamics of health service delivery and identify and resolve any challenges that may affect the study activities including data collection.

## Recruitment and training of data collectors

We recruited 4 graduate-level experienced research assistants and further train them for data collection. Research assistants were trained on rationale of the study, data collection tools, consent procedures and qualitative interviewing techniques include translating interview guides into the two main local languages (i.e Kasem and Nankani) spoken in the area for the purposes of consistency. Mock interviews were conducted during training, which was used to evaluate the performance of data collectors. Also, a pre-test was conducted at the end of the training, which had helped the study team to finalize the interview guides before the actual data collection. The pre-test data was not included in the study.

## Data collection techniques

Data collectors visited the selected districts and health facilities, introduced themselves to the district and facility leaders and conducted the interviews with health workers. Appointments were also booked with community members including venue, time and date before the FGDs and IDIs were conducted after informed consent was obtained.

Various COVID-19 pandemic measures including wearing of nose masks and the use of hand sanitizer were observed by both data collectors and study participants during the interviews. The interviews were recorded with consent of participants. At each stage of the data collection, data collectors were supervised to make sure that the data collection was done accurately. A total of 16 FGDs were conducted with mothers with children less than five years and opinion leaders. The number of people per each FGD varied between 9 and 12 participants. Also, 10 IDIs with parents whose children were affected with the sickle cell disease and 19 IDIs with health workers were conducted. Table 1 provides further information on category and number of interviews conducted.

## Data management and analysis techniques

All interviews were recorded and transcribed verbatim after repeatedly listening to the recordings. To ensure that qualitative principles of transcribing interviews were applied, different

**Table 1. Category and number of interviews conducted.**

| Interview type | Category | Number of interviews |
|---|---|---|
| IDIs | Community Health Nurses | 8 |
| | Medical Assistants | 4 |
| | District Public Health Nurses | 2 |
| | District Directors of Health | 2 |
| | Regional Public Health Nurse | 1 |
| | Regional Newborn and CHPS Coordinators | 2 |
| Total | | **19** |
| IDIs | Parents/caretakers of children with SCD | 10 |
| FGDs | Mothers with children under-five years | 8 |
| | Community Opinion Leaders | 8 |
| **Total** | | **16** |

people were engaged to transcribe the audio recordings. The transcripts were reviewed to correct typographical and grammatical errors to make them readable before data coding. A codebook containing main and sub-themes was developed to facilitate data coding. The codebook was developed using a combination of established categories based on the original research questions and other themes that emerged from the data. The transcripts were prepared and imported into QSR Nvivo 12 software and coded. The coding process involved a critical review of each transcript and coding of the data into emerging themes. Memos were created and attached to themes during data coding. These memos enabled coders to put down key findings and ideas from the data and also patterns that could be connected, compared or contrasted during the coding process.

Thematic content analysis was used to analyze the data. The process of thematic content analysis means reading through textual data, identifying themes, coding the themes and then interpreting the content of the themes [22]. The results were then presented as a narrative, supported with quotes from the data.

## Results

The results are presented to cover major thematic areas such as community awareness and perceptions of SCD, their views on integrating POCT to screen for SCD as well as barriers and facilitators to community acceptability of using POCT technology to screen for SCD in children under 5 years of age. Summary of main and sub-themes and quotations are described in Table 2.

### Awareness and perception of sickle cell disease

Views expressed by participants suggested that most of them were aware of the sickle cell disease. While some participants described it as sickness in the bones that caused frequent illness to children, others held that parents who had sickle cell disease could transfer it to their children. Participants reported that children with SCD would usually experience severe pain especially in the bones.

**Table 2. Summary of main and sub-themes and quotations.**

| Theme | Sub-theme | Quotes |
|---|---|---|
| **Awareness and perception of sickle cell disease** | Awareness of SCD | *I am aware of the disease. parents can transfer the SCD to their children if the parents have the condition.* **(FGD-under 5 mothers-Kasem-06)** |
| | Perception of SCD | *I see this sickness as very bad disease because you can lose someone you love as a result of the disease.* **(FGD-under 5 mothers-Kasem-12)** |
| **Views on SCD screening in children under five years old** | | *The screening will help parents to know their children's sickle cell disease status.* **(IDI-Parent with SCD patient-Kasem-02)** |
| **Barriers to uptake of POCT device to screen for SCD** | Discrimination among community members | *When nurses who are going to do the test start discriminating among parents, some people will bring their children.* **(FGD-opinion leader-Kasem-16)** |
| | Fear of being tested positive | *Fear of being tested positive can make people not to go for the screening.* **(FGD-under 5 mothers-Kasem-06)** |
| | Waiting time | *If the screening process is going to take a long time, I think that will be a challenge.* **(FGD-opinion leaders-Nankani-02)** |
| **Facilitators to community acceptability to the SCD screening** | Intensive health education | *We have to use the information centres to educate people before the work starts.* **(FGD with under 5 mothers-Kasem-06)** |
| | Positive attitude of health professionals | *If the health workers who are going to do the test have patience and also respect community members, it will encourage people to come for the screening.* **(FGD-opinion leaders- Kasem-04)** |

*When the child has the disease (referring to SCD), the child can feel pains in the body especially in the bones and that makes them fall sick all the time. (IDI-Parent-Kasem-04)*

*R8: This is sickness in the bones, which normally makes the child very weak and sick. (FGD-under 5 mothers-Nankani-03)*

*R9: I am aware of the disease because parents can transfer the sickle cell disease to their children if the parents have the condition. (FGD-under 5 mothers-Kasem-06)*

Nonetheless, a few participants reported having no knowledge about sickle cell disease. As one participant in the FGDs expressed it:

*R8: We know that the children fall sick but we do not know what is making them fall sick all the time. (FGD-under 5 mothers-Kasem-01)*

Most community members (105/148), representing about 71.0% described SCD as serious and potentially life-threatening condition affecting children. They noted that people could easily lose their love ones resulting from the disease.

*R5: I see this sickness as very bad disease because you can lose someone you love as a result oof the disease and that is very disturbing. (FGD-under 5 mothers-Kasem-12)*

A parent whose child was affected with sickle cell disease perceived the condition as spiritual affecting families.

*I think the disease (referring to SCD) is a spiritual disease affecting families. (IDI-Parent with SCD patient-Kasem-02)*

Most participants in the FGDs especially opinion leaders perceived that some community members would not get closer to a sickle cell disease patient/family if they knew the person had the condition.

*R5: Some people here, if they get to know that you have that disease, they will not get closer to you or your family; thinking that they may be infected. (FGD-opinion leaders-Nankani-02)*

*R3: People don't always want to get closer to such children. If the child fetches water, some people do not even want to drink the water. They normally think that if they get closer to the child or collect water from them and drink, they could be infected with the disease. (FGD-opinion leaders-Kasem-11)*

Other participants felt that married women who committed adultery and got pregnant, they could give birth to children who could have sickle cell conditions. Most of these views were expressed by mothers with children less than five years old as demonstrated in the following excerpts:

*R2: Others will also think that the woman has gone against tradition or taboos, which they called Kabunsi (A disease a woman gets when she commits adultery). (FGD-under 5 mothers-Nankani-10)*

*R4: Sometimes, the blame is put on the mother that she is the cause of everything. Some will say that the mother got the pregnancy outside the family so, the child is not part of the*

*family and that is why the child is having sickle cell disease. (FGD-under 5 mothers-Nan-kani-13)*

### Views on SCD screening in children under five years old

Of the 148 community members and health workers who were interviewed, 141 (95.2%) of them expressed positive views about the screening exercise. According to community members, the screening exercise could facilitate diagnosis of SCD in children for early management in the area.

*R3: When they bring this testing machine, it will really help us know the status of our children. This will also help the nurses to treat our children very well. (FGD-under 5 mothers-Kasem-07)*

*R6: It will be good to test the children because if a child is tested positive, the nurses will advise the parents on how to take care of the child. (FGD-opinion leaders-Nankani-02)*

*R: When you do the test, it will help parents to know their children's sickle cell disease status. (IDI-Parent with SCD patient-Kasem-02)*

Community members were very happy about the planned screening exercise and as a result, they expressed their readiness to support the research team for a successful exercise.

*R1: As chief, I am very happy about the program because you are coming to help my community to have good health and for that reason, we are ready to support the research team. (FGD-opinion leaders- Kasem-04)*

*R6: I am already happy because I see it as something that will bring a lot of help to most of us. This assistance will help us and I am already very happy even before the program will start. (FGD-under 5 mothers-Kasem-01)*

Health professionals also shared their views on the screening exercise in the following quotes when they were questioned:

*M: What do you think about the SCD screening exercise that the research team is going to introduce?*

*R: I think it is a good initiative that will go a long way to prevent some deaths among children under five years old. (IDI-public health nurse-02)*

*R: I think it will be a very good intervention that will help people and I will like to say that I am in support of the program. (IDI-director of health services-01)*

*R: I will say that the initiative is good because it will help us identify children with sickle cell and manage them as early as possible. (IDI-medical in-charge-03)*

The wish of a community health nurse was that the screening program be sustained to help community members. As she expressed it:

*I will like to add that the project is a good initiative so when we start, we are begging that it should be continued, it should be something that will come and stay to help the community. (IDI-community health nurse-11)*

## The use of community health nurses for the screening exercise

Almost all community members who were interviewed held that they did not have any problem if community health nurses were used for the screening exercise. According to participants, community members, especially women were used to working with the community health nurses because these nurses were already taking good care of their health needs at the primary healthcare facilities. Therefore, they believed that the community health nurses could handle them appropriately in the screening exercise, which could encourage mothers to patronise the program. Similar views were shared by both Kasem and Nankani participants on the issue in the discussions.

> R10: I think it is very good to use the community health nurses to do the test because they are there with us in the community and taking care of our health needs. (FGD-under 5 mothers-Kasem-06)

> R4: I think the nurses (referring to community health nurses) are the appropriate people to be used for the test because they have been trained to do that. (FGD-under 5 mothers-Nankani-13)

Similar views were also expressed by health workers when they were questioned on the possibility of community health nurses adding the SCD screening exercise to their routine duties at the health facility level.

> M: Is it possible for community health nurses to routinely use this device to screen for SCD in children?

> R: I think the community health nurses mostly deal with the children because they do a lot of things such as immunization, outreaches, home visit, etc. So, using the community health nurses, for me, is the best. (IDI-community health nurse-12)

> R: Yes, it is very possible because it is part of what we do already. We attend to children so we will just add it to our programs. (IDI-community health nurse-01)

> R: As a community health nurse, we work with children under five years. So, when the mothers come for weighing, we will explain to them why we are doing the SCD screening. (IDI-old community health nurse-05)

## Barriers to uptake of POCT device to screen for SCD

Various factors that could affect acceptability and uptake of the screening for SCD in children under five years of age in the study area were reported by participants. For instance, discrimination among community members by screening officers was highly reported by community members in the interviews as key factor that could affect interest and patronage of the SCD screening exercise as exemplified in the following extracts:

> R6: The issue is that when those who dress and look rich come and the nurses attend to them first before others, that could be a problem. (FGD-under 5 mothers-Kasem-07)

> R5: Those of us who are privileged to attend school small would be called first leaving those who are not well educated. When that happens, it could affect patronage of the exercise. (FGD-under 5 mothers-Nankani-10)

> R3: When the exercise starts and nurses who are going to do the test start discriminating among community members, some people will lose interest in bringing their children for the test. (FGD-opinion leader-Kasem-16)

According to some community members, fear of being tested positive for SCD resulting to stigmatization by other community members could also affect patronage of the screening exercise.

*R8: Fears of being tested positive can make people not to go for the screening. (FGD-under 5 mothers-Kasem-06)*

*R6: When a person is tested positive and the community members are stigmatizing the person, I think some parents will not agree to send their children for the screening. (FGD-opinion leaders -Nankani-05)*

Negative health worker attitude linked with the issue of maintaining confidentiality were also reported as important factors that could have a negative influence on patronage and uptake of the SCD screening exercise by parents. This was largely reported by health professionals as illustrated in the following quotes:

*If we (referring to health workers) turn to disclose their information to other people, community members will not trust us and that could discourage parents from coming for the exercise. (IDI-community health nurse-02)*

*The attitude of some health workers could be a problem. I think if health workers who will be engaged in the exercise do not behave well towards community members and parents, it may affect their interest in the exercise. (IDI-public health nurse-02)*

*R4: If health workers who are going to screen our children do not know how to talk to people, it can make parents feel reluctant to send their children for the test. (FGD-opinion leaders-Nankani-05)*

Waiting for so long to go through the screening exercise was also reported as potential factor that could discourage some parents to bring their children for the screening exercise.

*R5: If the screening exercise is going to take a long time, I think that will be a challenge. (FGD-opinion leaders-Nankani-02)*

A few participants also perceived that lack of personal interest and understanding resulting from poor sensitization and communication on the importance of the SCD screening exercise, coupled with the fact that the screening exercise was new to community members could affect their patronage.

*R2: The challenge will be that, most people will not come out and test if they do not understand the benefits and why they should bring their children to be screened. (FGD-under 5 mothers-Kasem-01)*

*R4: . . .because it is a new thing, it will be difficult for some people to agree and bring their children for testing. (FGD-opinion leaders-Nankani-02)*

*I think sometimes the community members are difficult and some of them even though you are helping them, they will not show any interest. (IDI-community health nurse-02)*

A good number of community members and health workers perceived that traditional and religious beliefs could also affect patronage of the SCD screening exercise in the area as demonstrated in the following quotes:

*R1: Some traditional believers do not like going to hospital. Believers in some churches such as deeper life people do not go to hospital when they are sick. So, this testing, they will not agree and come. (FGD-under 5 mothers-Nankani-10)*

*R3: Others believe in their gods to heal them than going to the hospital and such people will not come for the test. (FGD-opinion leaders-Kasem-08)*

*Some people may not agree for their children to be tested because the belief is that the ancestors will heal the child. (IDI-Public Health Nurse-04)*

*Traditionally, some people do not allow their family members to be tested in the hospital because they believe in the herbalists. (IDI-Medical In-charge-09)*

## Facilitators to community acceptability to the SCD screening

Participants recommended various strategies that could promote community acceptability and uptake of POCT for SCD in children under five years old in the study area. These included health education, short duration of screening processes, positive attitude of health workers and screening health workers not being biased.

## Intensive health education

About 78.4% (110/148) of participants mentioned community education focusing on the importance for the SCD screening exercise in children under five years old could enhance community patronage and uptake of the exercise by parents in the area.

*R2: If community members could be educated more on the disease and the screening exercise, it will help people to understand the importance for the screening and the need for them to allow their children to take part. (FGD-opinion leaders -Nankani-05)*

*When the nurses are doing the testing, they should continue telling parents the importance of the exercise especially to the children. (IDI-Parent with SCD patient-Nankani-01)*

*When we educate parents very well to understand the need for the exercise, they will embrace it and bring their children for the screening. (IDI-29yr community health nurse-12)*

Evidence based education touching on previous works done by the research centre in the area to improve health of community members was also recommended as an important strategy that could facilitate community acceptability and uptake of the SCD screening exercise.

*R6: I think if they are going to educate people, the research team should start with similar things they have done in the past to solved health-related issues in the area and I think this will help people to show interest. (FGD-under 5 mothers-Nankani-03)*

Similarly, the use of community information centres and radio stations to educate and also make announcements about the screening exercise could improve awareness and uptake of the exercise according to participants.

*R1: We have to use the information centres to make announcements and also educate people before the work starts. (FGD with under 5 mothers-Kasem-06)*

*Announcements should be made in the various communities using community information centres so that people will be aware of the exercise and come out for it. (IDI-Parent with SCD Patient-Nankani-07)*

Involvement of chiefs and other community leaders in the program was also recommended to improve community acceptability of the SCD screening exercise in the area. They explained that people had respect for community leaders and if such people were made to talk to other community members in the area, it could improve trust and patronage of the exercise.

*They should involve community leaders in the program to prove to the community members that what the research team is going to do is good and will benefit them. When the people see their own community leaders among the research team, they will come out for the test. (IDI--Parent with SCD Patient-Kasem-06)*

The use of community health volunteers to educate people at the community level was also recommended to create awareness about the SCD screening exercise in the area.

*R4: What I will say is that they (referring to research team) should try and get community volunteers and train them to educate people on sickle cell disease and help organize them for the screening. (FGD-opinion leaders-Nankani-02)*

*I think we have to get volunteers to talk to community members about the exercise and that will enable many people to be aware of the program and come out with their children for the screening. (IDI-Medical In-charge-09)*

## Short duration of the screening process

Also, about 74.5% of participants were of the view that short duration of the screening process could encourage many parents to take part in the exercise.

*R7: If the testing process will not take a longer time, people will patronise it. (FGD-Opinion leaders-Kasem-08)*

*R2: What I will like to add is that if the testing process is fast, it will encourage more people to go and test. (FGD-opinion leaders -Nankani-05)*

## Positive attitude of health professionals

About 70.0% (103/148) of participants also held that positive attitude of health professionals and screening nurses not being biased could encourage community members especially parents to send their children to the screening health facility for the SCD screening.

*R8: They (referring to screening nurses) should not be biased and when they are not biased, it will encourage people to come and test. (FGD- under 5 mothers-Nankani-13)*

*R7: If the health workers who are going to do the test have patience and also respect community members, it will encourage people to come for the screening exercise. (FGD-opinion leaders- Kasem-04)*

## Discussion

The current study explored the feasibility and community acceptability of integrating Gazelle POCT technology to screen for SCD in children under 5 years of age in primary healthcare facilities to initiate care and referral pathways for early and comprehensive management. Further, the study explored views of participants on barriers that could affect patronage of the screening exercise and their ideas on appropriate strategies to promote community acceptability and uptake of SCD screening in the area.

Participants described sickle cell disease as serious and potentially life-threatening condition affecting children in the study area. They noted that people could easily lose their children or loved ones resulting from sickle cell disease. This perception by community members could be because of their experiences on the health difficulties people especially, children who have sickle cell disease go through in the study communities as pointed out by parents/caretakers of these children in the study. It is reported earlier that sickled cells disease could obstruct blood flow to specific organs, leading to stroke, acute chest syndrome, organ damage and in many cases premature death [23]. As a result, public health interventions are recommended to create awareness about this dangerous and life-threatening condition as well as strategies to reduce the prevalence of the disease. Sickle cell disease has also been perceived by participants in our study as spiritual and punishment to families with the condition. This perception could be attributed to lack of information about sickle cell disease by community members in the area as noted by earlier studies that lay people were not aware that SCD is a hereditary condition [24,25].

Both community members and health professionals in our study saw the need for sickle cell disease screening in children, noting that this laudable initiative could facilitate diagnosis of SCD in children for early management. The positive views expressed by participants, their eagerness and readiness to embrace the exercise could be that such an important health program has never been implemented in the area as reported by health professionals who were interviewed in the study. Further, the positive views expressed by participants could also be as a result of the benefits such as free screening and medical care for their children. This is consistent with earlier studies reporting that similar benefits motivated people to take part in health intervention programs [26,27].

Despite positive views shared by participants in our study, the findings point to key barriers to community acceptability and uptake of the SCD screening program in the area. For instance, discrimination, fear of being tested positive, stigmatization, negative health worker attitude linked with issues of maintaining confidentiality have all been reported by the majority of participants as key factors that could negatively affect acceptability and uptake of the SCD screening exercise by community members and parents. It is demonstrated that negative health worker attitude towards clients such as physical abuse, lack of regard for privacy, frightening attitudes and poor communication have been reported to undermine patronage of health interventions to improve health outcomes at the community level [28,29]. It is not therefore surprising that participants in the current study reported unpleasant and undesirable behaviors of health professionals linked with issues of discrimination and their inability to maintain confidentiality as important factors, which could affect patronage of the SCD screening exercise in the current study. Thus, the need for the SCD screening team to take step towards addressing the issue of health worker negative behavior to promote community interest and patronage of the SCD screening exercise by parents is highly recommended.

In addition, individual attitude and interest are key to acceptability of health programs at the community level. This means that if parents or community members are not convinced about the need for the SCD screening exercise resulting from inadequate community

engagement and education, it could affect their support, patronage and involvement in the exercise as pointed out by Sekhon et al., in their theoretical framework on acceptability of healthcare interventions. The framework explains that prior to participating in health interventions, individual attitude, burden associated with participating in the intervention and the efforts required to participate could all influence patronage and acceptability [21]. Indeed, these factors came up very strongly in the current study as key factors that could have a negative effect on uptake of SCD screening exercise.

However, both community members and health professionals suggested various strategies such as intensive health education, short duration of screening processes and screening health workers not being biased could promote community acceptability of SCD screening in the area. Positive health worker attitude toward clients at health facility level has been reported in current literature to improve patronage of health interventions [30]. In this study, most participants believed that effective and continuous community education and sensitization could promote community acceptability and uptake of the SCD screening exercise. More importantly, providing evidence-based education on similar exercises carried out by the Navrongo Health Research Centre in the past to improve health outcomes in the area and announcements on community media stations have been highly recommended to create awareness and understanding about the benefits and need for parents to accept the SCD screening exercise. Again, these findings support the theoretical framework on acceptability of healthcare interventions used in this study. According to the framework, the extent to which people understand health interventions, benefits and perceived effectiveness of these interventions could have a positive influence on community acceptability of such interventions, in this case the SCD screening exercise [21].

Community engagement involves interactive relationship between researchers and the community where people are seen as partners in the research process. Focusing on the need to implement health interventions to improve health outcomes of community members especially children through community education could provide valuable input in identifying ways to improve community understanding and acceptability of such interventions [31,32]. This suggests that effective engagement with key stakeholders such as chiefs, elders, opinion leaders, parents and health professionals at the initial stages of the SCD screening exercise is recommended to improve acceptability and uptake of the exercise.

## Strengths and limitations

A strength of this research is the selection and representation of men and women from the main local languages speaking in the area in the FGDs, as well as parents whose children have been affected with the sickle cell disease and health professionals. This allowed us to obtain a diversity of information about the screening services, which had informed the implementation design for using POCT to screen for SCD in primary care settings in the area. The main limitation was that the interviews with community members were conducted in the main local languages of the study area, tape recorded, transcribed and translated into English. It is possible that some statements made in the local languages may have lost their original meaning in the English translation. However, the interviews were transcribed by graduate research officers who are native speakers with experience in transcribing qualitative interviews. Any loss of meaning during the translation was minimized and did not affect the findings of the study.

## Conclusion

Majority (over 95%) of the participants viewed screening for SCD in children as very important. However, views expressed by participants suggest that health education, short duration of

SCD screening process, and professionalism of health workers in keeping patients' information confidential could improve community acceptability. These findings have informed the implementation design for using POCT device to screen for SCD in children under five years of age in primary healthcare facilities in rural Northern Ghana.

## Supporting information

**S1 Data. Community members.**
(ZIP)

**S2 Data. Health workers.**
(ZIP)

## Acknowledgments

The authors would like to express their profound gratitude to all the study participants for sharing their views with the research team. We are also grateful to the research assistants (i.e Miss. Nadia Fagenatera Anuseh and Mr. Stephen Azalia) who supported the research team during data collection.

## Author Contributions

**Conceptualization:** Samuel T. Chatio, Nicola Lister, Raymond Aborigo, Patrick Ansah, Isaac Odame.

**Data curation:** Samuel T. Chatio.

**Formal analysis:** Samuel T. Chatio, Enyonam Duah.

**Funding acquisition:** Nicola Lister, Olufolake Egbujo, Kwaku Marfo, Isaac Odame.

**Investigation:** Samuel T. Chatio, Lucy O. Ababio, Raymond Aborigo, Patrick Ansah, Isaac Odame.

**Methodology:** Samuel T. Chatio, Enyonam Duah, Lucy O. Ababio, Raymond Aborigo, Patrick Ansah, Isaac Odame.

**Project administration:** Olufolake Egbujo, Patrick Ansah.

**Resources:** Olufolake Egbujo, Kwaku Marfo.

**Software:** Samuel T. Chatio.

**Supervision:** Enyonam Duah, Lucy O. Ababio, Raymond Aborigo.

**Visualization:** Nicola Lister, Kwaku Marfo, Patrick Ansah.

**Writing – original draft:** Samuel T. Chatio.

**Writing – review & editing:** Enyonam Duah, Lucy O. Ababio, Nicola Lister, Olufolake Egbujo, Kwaku Marfo, Raymond Aborigo, Patrick Ansah, Isaac Odame.

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
