## [Decision Letter · Decision Letter 0]

21 Feb 2024

PONE-D-23-12189

Barriers and facilitators to community acceptability of integrating point-of-care testing to screen for sickle cell disease in children in primary healthcare settings in rural Northern Ghana.

PLOS ONE

Dear Dr. Chatio,

Thank you for submitting your manuscript to PLOS ONE. After careful consideration, we feel that it has merit but does not fully meet PLOS ONE’s publication criteria as it currently stands. Therefore, we invite you to submit a revised version of the manuscript that addresses the points raised during the review process.

We look forward to receiving your revised manuscript.

Kind regards,

Enoch Aninagyei, PhD

Academic Editor

PLOS ONE

Journal Requirements:

   "This study was part of a research collaboration grant provided by Novartis Pharma AG (grant ID: CGZX411A12402R) to Hospital for Sick Children and Navrongo Health Research Center. "

3. In the online submission form you indicate that your data is not available for proprietary reasons and have provided a contact point for accessing this data. Please note that your current contact point is a co-author on this manuscript. According to our Data Policy, the contact point must not be an author on the manuscript and must be an institutional contact, ideally not an individual. Please revise your data statement to a non-author institutional point of contact, such as a data access or ethics committee, and send this to us via return email. Please also include contact information for the third party organization, and please include the full citation of where the data can be found.

Reviewers' comments:

Reviewer's Responses to Questions

**Comments to the Author**

1. Is the manuscript technically sound, and do the data support the conclusions?

Reviewer #1: Yes

Reviewer #2: Partly

2. Has the statistical analysis been performed appropriately and rigorously? 

Reviewer #1: N/A

Reviewer #2: Yes

3. Have the authors made all data underlying the findings in their manuscript fully available?

Reviewer #1: No

Reviewer #2: No

4. Is the manuscript presented in an intelligible fashion and written in standard English?

Reviewer #1: Yes

Reviewer #2: Yes

5. Review Comments to the Author

Reviewer #1: Date: 10/06/2023

Reviewer: Desmond Kuupiel (MPH, Ph.D.)

General comment

Thank you for giving me the opportunity to review this manuscript entitled “Barriers and facilitators to community acceptability of integrating point-of-care testing to screen for sickle cell disease in children in primary healthcare settings in rural Northern Ghana”. I find this study very useful, interesting, and well written. However, I recommend the authors address the few comments outlined below prior to its acceptance for publication.

Minor

1. I noticed that this study involved only two districts in the Upper Region. So, I suggest the title be revised to reflect the true study region i.e., Upper East Region and not Northern Ghana. Northern Ghana has five administrative regions.

2. I understand that the study participants were purposively selected, but how the communities and the health facilities (e.g., CHPS and health centres) from each zone were sampled has not been captured. I suggest you revise to clearly capture this.

3. It is best to let readers know the denominator of each percentage where necessary. For example, “Most participants (70.9%) described SCD as …”; “About 78.4% of participants mentioned …”

Major

1. Gazelle POCT technology to screen for sickle cell disease has been mentioned severally, and yet, no literature on it has been provided in this manuscript. I think it will be fair to let potential readers know what Gazelle POCT technology is. Its accuracy, availability, affordability, suitability for at the primary care level compared to others, etc. Perhaps, this can be captured in the background section.

2. This study has several strengths and weaknesses, yet none of them has been discussed. Please included these as well as recommendations in your discussion.

I hope this helps.

Thank you.

Reviewer #2: Barriers and facilitators to community acceptability of integrating point-of-care testing to

screen for sickle cell disease in children in primary healthcare settings in rural Northern

Ghana.

This is an interesting manuscript. The researchers undertook a primarily qualitative study to determine the attitudes of health care workers and community members to a newborn screening program.

The abstract is the first place where it is asserted that “Point-of-care testing (POCT) for SCD screening can easily be integrated into existing immunization programs”. This assertion has not been justified. It is clear from the manuscript that extensive training and changing of attitudes of health care workers and community members would be required. Significant resource inputs would be required. So, whereas it may not be as difficult as other approaches, it is by no means easy.

Introduction

Line 81/82 The sentence “Also, pneumococcal infections and malaria-related severe anemia are among the commonest causes of SCD-related mortality [5].” does not belong in this paragraph.

Line 88/89 The statement “Several pilot initiatives in Africa have demonstrated that newborn screening coupled with early intervention for SCD is feasible and can yield outcomes commensurate with those achieved in high-income countries.” requires references.

Line 93/94 The phrase “While pilot initiatives in Africa have demonstrated that neonatal screening coupled with early

intervention reduces SCD-related morbidity and mortality,” is repetitive, having occurred in the previous paragraph. It can be shortened.

Methods

Details should be added about the SCD POCT to which the community was exposed. Was their knowledge of POCT with that system theoretical or did they actually experience testing? There are responses to the “short duration of screening”.

The instruments are not described or provided. Appendices with the prompts used for focus groups and questions for in depth interviews would be helpful.

For the instruments, information should be provided on any pre-testing, piloting and assessment of validity in the context.

Were there 16 focus groups? How many individuals participated in each focus group?

Results

The results may be more digestible in a table where the quotes are presented in themes.

Discussion

Much of the discussion is repeating the results without much comparison and contextualization within existing literature. Where there are gaps in the literature, this can be stated.

References

Reference 3 doesn’t seem to be the appropriate citation.

6. PLOS authors have the option to publish the peer review history of their article (what does this mean?). If published, this will include your full peer review and any attached files.

Reviewer #1: **Yes: **Dr. Desmond Kuupiel

Reviewer #2: No

---

## [Author Response · Author response to Decision Letter 0]

13 Mar 2024

Manuscript ID: PONE-D-23-12189

Title: Barriers and facilitators to community acceptability of integrating point-of-care testing to screen for sickle cell disease in children in primary healthcare settings in rural Upper East Region of Northern Ghana

Response to reviewers' comments:

General comment

Thank you for giving me the opportunity to review this manuscript entitled “Barriers and facilitators to community acceptability of integrating point-of-care testing to screen for sickle cell disease in children in primary healthcare settings in rural Northern Ghana”. I find this study very useful, interesting, and well written. However, I recommend the authors address the few comments outlined below prior to its acceptance for publication.

Response: We thank the reviewers for the positive comment.

Minor

1. I noticed that this study involved only two districts in the Upper East Region. So, I suggest the title be revised to reflect the true study region i.e., Upper East Region and not Northern Ghana. Northern Ghana has five administrative regions.

Response: We thank the reviewers for this suggestion. The title has accordingly been revised to read as follows: Barriers and facilitators to community acceptability of integrating point-of-care testing to screen for sickle cell disease in children in primary healthcare settings in rural Upper East Region of Northern Ghana (Kindly refer to the title page of the revised manuscript for this revision) 

2. I understand that the study participants were purposively selected, but how the communities and the health facilities (e.g., CHPS and health centres) from each zone were sampled has not been captured. I suggest you revise to clearly capture this.

Response: The sampling paragraph has been revised to capture the sampling procedures for health facilities, health workers and communities. 

Selection of health facilities and health workers

Concerning the sampling of health facilities and health workers, the sentence has been revised to read as follows: Using these demarcations, we purposively selected 4 high delivery Community-based Health Planning and Services (CHPS), clinics and health centres in each of the language groupings for the study. Thus, health workers such as medical assistants and community health nurses in the study health facilities were purposively selected and invited for the interviews. The other interviews with Directors, Public health nurses and newborn and CHPS coordinators were conducted at the district and regional levels respectively. These category of health professionals were also purposively selected and included in the study due to their role in policy implementation at the district and regional levels (kindly refer to the sampling procedures, page 8, second paragraph, lines 6 to 13 for the revised)

Selection of communities

Similarly, purposive sampling technique was used to select four communities (i.e. Wuru, Chiana-Asunia representing the Kasem speaking area and Mirigu, Biu representing the Nankani speaking area) within the catchment areas of the study health facilities for the FGDs with community members (kindly refer to page 8, last paragraph, lines 1 to 3 for the revised)

3. It is best to let readers know the denominator of each percentage where necessary. For example, “Most participants (70.9%) described SCD as …”; “About 78.4% of participants mentioned …”

Response: Denominators were added to the percentages used in the revised manuscript. For example, most community members (105/148), representing about 71.0% described SCD as serious and potentially life-threatening condition affecting children (Please, refer to page 12, third paragraph, line one for this addition)

Of the 148 community members and health workers who were interviewed, 141 (95.2%) of them expressed positive views about the screening exercise (Please, refer to page 14, first paragraph, line 1 for this information)

About 78.4% (110/148) of participants mentioned community education focusing on the importance for the SCD screening exercise in children under five years old could enhance community patronage and uptake of the exercise by parents in the area (Please, refer to second paragraph, line 1 on page 20 for this addition)

About 70.0% (103/148) of participants also held that positive attitude of health professionals and screening nurses not being biased could encourage community members especially parents to send their children to the screening health facility for the SCD screening (Kindly refer to page 22, first line under the health worker attitude for this addition)

Major

1. Gazelle POCT technology to screen for sickle cell disease has been mentioned severally, and yet, no literature on it has been provided in this manuscript. I think it will be fair to let potential readers know what Gazelle POCT technology is. Its accuracy, availability, affordability, suitability for at the primary care level compared to others, etc. Perhaps, this can be captured in the background section.

Response: A paragraph on information about Gazelle technology has been added to the revised manuscript as follows: Gazelle is a single-use cartridge-based test that enables rapid, affordable, accurate diagnosis for both SCD and sickle cell trait at the point of care. It is a cellulose acetate-based microchip electrophoresis system within a portable instrument that applies the principle of standard electrophoresis method with inbuilt analysis software, electronic data storage, and wireless data transmission capabilities. Subjects with SCD can be identified at 100% sensitivity and specificity from normal subjects and subjects with sickle cell trait. Additionally, subjects with sickle trait can be identified at 98.1% sensitivity and 97.0% specificity from normal subjects. The device is operated by rechargeable lithium batteries that can test all day on a single charge. The device also carries advanced features such as WiFi, GPS, and Bluetooth that allows connectivity for easy tracking of samples. (Please, refer to the second paragraph, page 6 of the revised manuscript for this addition)

2. This study has several strengths and weaknesses, yet none of them has been discussed. Please included these as well as recommendations in your discussion.

Response: We thank the reviewers for this comment. A statement on strengths and limitation of the study have been added to the revised manuscript as follows: “A strength of this research is the selection and representation of men and women from the main local languages speaking in the area in the FGDs, as well as parents whose children have been affected with the sickle cell disease and health professionals. This allowed us to obtain a diversity of information about the screening services, which had informed the implementation design for using POCT to screen for SCD in primary care settings in the area. The main limitation was that the interviews with community members were conducted in the main local languages of the study area, tape recorded, transcribed and translated into English. It is possible that some statements made in the local languages may have lost their original meaning in the English translation. However, the interviews were transcribed by graduate research officers who are native speakers with experience in transcribing qualitative interviews. Any loss of meaning during the translation was minimised and did not affect the findings of the study” (Please, refer to last paragraph, page 26 for this addition)

Also, recommendations have been added in the discussion where necessary (Kindly refer to the discussion section of the revised manuscript)

Reviewer #2: Barriers and facilitators to community acceptability of integrating point-of-care testing to screen for sickle cell disease in children in primary healthcare settings in rural Northern Ghana.

This is an interesting manuscript. The researchers undertook a primarily qualitative study to determine the attitudes of health care workers and community members to a newborn screening program.

Response: We thank the reviewer for the positive comment. 

The abstract is the first place where it is asserted that “Point-of-care testing (POCT) for SCD screening can easily be integrated into existing immunization programs”. This assertion has not been justified. It is clear from the manuscript that extensive training and changing of attitudes of health care workers and community members would be required. Significant resource inputs would be required. So, whereas it may not be as difficult as other approaches, it is by no means easy.

Response: We thank the reviewer for this comment, we have therefore taken off the word easily from the sentence as recommended by the reviewer. 

Introduction

Line 81/82 The sentence “Also, pneumococcal infections and malaria-related severe anemia are among the commonest causes of SCD-related mortality [5].” does not belong in this paragraph.

Response: The sentence has been deleted from the paragraph. 

Line 88/89 The statement “Several pilot initiatives in Africa have demonstrated that newborn screening coupled with early intervention for SCD is feasible and can yield outcomes commensurate with those achieved in high-income countries.” requires references.

Response: The sentence has been referenced as follows: Oluwole EO., Adeyemo TA., Osanyin GE., Odukoya OO., Kanki PJ.,. Afolabi BB (2020) Feasibility and acceptability of early infant screening for sickle cell disease in Lagos, Nigeria-A pilot study. PLoS ONE; 15(12): e0242861 (Kindly refer to the reference number 12) 

Line 93/94 The phrase “While pilot initiatives in Africa have demonstrated that neonatal screening coupled with early

intervention reduces SCD-related morbidity and mortality,” is repetitive, having occurred in the previous paragraph. It can be shortened.

Response: The sentence has been revised based on the reviewer’s suggestion. 

Methods

Details should be added about the SCD POCT to which the community was exposed. Was their knowledge of POCT with that system theoretical or did they actually experience testing? There are responses to the “short duration of screening”.

Response: Thanks for the comments. Regarding the views of community members about the short duration of the screening was not about their experience regarding the gazelle POCT device. They were referring to the whole screening process and the time it would take for a participant to go through the process. They felt, that if the screening process would take a longer time, it would discourage some community members to patronise the exercise.

The instruments are not described or provided. Appendices with the prompts used for focus groups and questions for in depth interviews would be helpful.

Response: The interview guides have been attached.

For the instruments, information should be provided on any pre-testing, piloting and assessment of validity in the context.

Response: A sentence on pre-test has been added to the revised manuscript to read as follows: Also, a pre-test was conducted at the end of the training, which had helped the study team to finalize the interview guides before the actual data collection. The pre-test data was not included in the study. (Kindly refer to page 9, last paragraph, lines 6 to 8 for this information)

Were there 16 focus groups? How many individuals participated in each focus group?

Response: A sentence on the number of participants per FGD has been added to read as follows: The number of people per each FGD varied between 9 and 12 participants (Please, refer to page 10, second paragraph, line 6 for this addition)

Results

The results may be more digestible in a table where the quotes are presented in themes.

Response: A table has containing summary of main and sub-themes as well as quotes have been added as shown below (Kindly refer to page 33 for this table)

Discussion

Much of the discussion is repeating the results without much comparison and contextualization within existing literature. Where there are gaps in the literature, this can be stated.

Response: The discussion section has been revised to take care of this concern as well as recommendations where appropriate. (Kindly refer to the discussion section for these revisions)

References

Reference 3 doesn’t seem to be the appropriate citation.

Response: The reference has been deleted 

The authors would want to thank the academic editor and reviewers for your valuable comments to improve the overall presentation of this manuscript. Therefore, we have addressed all comments to the best of our ability and hope that our manuscript will be considered for publication.

---

## [Editor Report · Decision Letter 1]

26 Apr 2024

Barriers and facilitators to community acceptability of integrating point-of-care testing to screen for sickle cell disease in children in primary healthcare settings in rural Upper East Region of Northern Ghana.

PONE-D-23-12189R1

Dear Dr. Chatio,

We’re pleased to inform you that your manuscript has been judged scientifically suitable for publication and will be formally accepted for publication once it meets all outstanding technical requirements.

Kind regards,

Enoch Aninagyei, PhD

Academic Editor

PLOS ONE
---

## [Editor Report · Acceptance letter]

7 May 2024

PONE-D-23-12189R1 

PLOS ONE

Dear Dr. Chatio, 

I'm pleased to inform you that your manuscript has been deemed suitable for publication in PLOS ONE. Congratulations! Your manuscript is now being handed over to our production team.

Kind regards, 

on behalf of

Dr Enoch Aninagyei 

Academic Editor

PLOS ONE